# Advocacy for Responsible Antibiotic Production and Use

**DOI:** 10.3390/antibiotics11070980

**Published:** 2022-07-20

**Authors:** Véronique Mondain, Nicolas Retur, Benjamin Bertrand, Florence Lieutier-Colas, Philippe Carenco, Sylvain Diamantis

**Affiliations:** 1Centre Hospitalo-Universitaire de Nice, Service des Maladies Infectieuses et Tropicales, Hôpital Archet 1, 06202 Nice, France; 2Centre Hospitalo-Universitaire de Nice, Pharmacie, Hôpital Archet 1, 06202 Nice, France; retur.n@chu-nice.fr; 3Centre Hospitalier de Grasse, Pharmacie, 06130 Grasse, France; benjamin.bertrand@gmail.com; 4AntibioEst, Regional Antibiotic Therapy Centre, Nancy University Hospital, Brabois Hospital, 54511 Vandoeuvre-les-Nancy, France; f.lieutier@chru-nancy.fr; 5Centre Hospitalier de Hyères, Hygiène, 83407 Hyères, France; pcarenco@ch-hyeres.fr; 6Centre Hospitalier de Melun, Infectiologie, 77011 Melun, France; sylvain.diamantis@ghsif.fr

**Keywords:** antimicrobial resistance, environmental impact, antibiotic use in livestock, antibiotic production

## Abstract

Antibiotic-resistant bacteria have become one of humankind’s major challenges, as testified by the UN’s Call to Action on Antimicrobial Resistance in 2021. Our knowledge of the underlying processes of antibiotic resistance is steadily improving. Beyond the inappropriate use of antimicrobials in human medicine, other causes have been identified, raising ethical issues and requiring an approach to the problem from a “One Health” perspective. Indeed, it is now clear that the two main issues regarding the subject of antibiotics are their misuse in the global food industry and their method of production, both leading to the emergence and spread of bacterial resistance.

## 1. Antibiotic Use in Animals

Since the 1950’s, following their initial authorization by the Food and Drug Administration, antibiotics have been used as a food supplement to promote weight gain in farmed animals. Today, 70% of all antibiotics consumed worldwide are absorbed by livestock, mainly to boost production and profits [1,2,3,4]. The permanent addition of a small daily dose of antibiotics to animal fodder results in approximately a 2% weight gain due to the resulting altered composition of the microbiota and an increased calorie uptake. Animals discharge these antibiotics in the soil via their feces and urine, contaminating natural environments. They also harbour multi-resistant bacteria that have been selected in their gut by the antibiotics they absorb, and hence pass them on to human populations who consume meat and other animal products. Current data collected by the ECDC, which conducts surveillance over 3rd generation cephalosporin-resistant Enterobacteriaceae in samples of European meat, show that 50% of samples are contaminated, with wide variations in this between countries [5]. This also applies to fish, although there are fewer studies on whether the samples originate from fish farms or wild species (studies conducted in Chinese markets [6], Korean farms [7], or Tanzanian lakes [8]) (Figure 1).

While Europe took steps to forbid the administration of antibiotics in growth promotion in 2006 [9], and the French eco-antibio programme has reduced antibiotic use from 25% to 37% [10], consumption remains significant due to high-density farming. The resulting potential for the extremely rapid spread of infection requires preventive treatment of the entire flock if only a few animals become sick (metaphylaxis). Outside Europe, most countries still use antibiotics to boost growth (USA, China, South America, Africa, etc.) and are forecast to increase their use in livestock breeding [11]. Imports, favoured by international trade agreements, are thus a source of high-risk products.

Although precisely assessing the impact of antibiotic use in livestock on bacterial resistance in human populations is difficult, evidence is increasing, and many authors insist on the need to reorientate antibiotic use [12,13].

## 2. Antibiotic Manufacturing

Since the 1990’s, most major pharmaceutical groups have relocated their production to emerging countries, mainly China and India, due to highly favourable economic conditions, i.e., cheap labour, and the lack of environmental constraints such as were beginning to appear in Western countries. Joachim Larsson was a whistle-blower in 2007 when he measured ciprofloxacin concentrations of 31 milligrams per litre in the effluent from a wastewater treatment plant serving drug manufacturing plants in India [14]. Since then, the NGO Changing Markets should be commended for its numerous reports showing the resulting environmental and health-related damage (Figure 1) [15]. Polluted lakes and rivers in countries where domestic sewage systems are rudimentary lead to contact between large quantities of enteric bacteria and antibiotic compounds, thus selecting multi-drug resistant (MDR) bacteria. Many studies currently report significant concentrations of quinolones, sulfonamides, and anti-fungal compounds in rivers in the region of Hyderabad, in central India, which is one of the largest bulk-drug manufacturing hubs. These lakes and rivers also contain large numbers of MDR bacteria and resistance genes such as ESBL or carbapenemase (NDM1, KPC…) [16,17], which exert a high toll on local populations through the food chain and water supply. At least 50,000 children are reported to die each year in this area due to untreatable MDR bacterial diarrhoea [18]. In a study on sepsis in neonates in North India, up to 90% of bacteria were MDR [19]. International trade and travel (tourism and medical tourism, India’s health resources currently attracting a large number of patients from developing countries) and the massive increase in air travel all facilitate the spread of resistance across the globe [20]. A review of fecal carriage rates of ESBL-producing Enterobacteriaceae shows the extent of dissemination of such strains [21]. In addition to this human-related spread, animal migration, ocean currents, and continental water flows contribute to the creation of a gene pool of antibiotic resistance hitherto scarcely explored via a “One Health” approach and which reaches the most remote planetary ecosystems such as the polar regions [22,23,24].

As Western countries, who have outsourced our drug manufacturing plants for profitability, we are now faced with the consequences of this choice, which raises ethical and health-related issues, along with strategic concerns in terms of supply and industrial autonomy. This situation calls for urgent attention and reaction. China launched its Blue Sky initiative in 2018 to improve air quality, and its environmental policy appears to have taken positive, though limited, steps in terms of tackling air pollution [25], although no definite information has been revealed regarding effluent pollution. However, India’s new “Pharma City” project seeks to obtain environmental clearance, yet this will still require impact assessments [10].

Purchasing countries have the leverage to require such supervision but also to assist these countries in their transition towards more virtuous practices while providing them with the capacity to produce without harming their population and their environment.

### 2.1. The Opacity of the Antibiotic Production Chain

When the medical director of a pharmaceutical company presents you with a new compound, your query regarding its geographical origin is unlikely to get a reply. A working group of physicians and pharmacists questioned ten major pharmaceutical companies, among which were producers of generic compounds. Six of them declined to answer, and four refused on the grounds of industrial secrecy [26]. All were surprised by the question, stating it was the first time that they had been asked. However, this should become a crucial concern at a time when drug availability is no longer guaranteed (868 notifications of supply tensions or disruptions in 2018, i.e., twenty-fold higher than in 2008) [27]. Legally, there is also a duty to declare the product’s origin to ensure the continuity of supply, and the French National Agency for the safety of medicines and health products (ANSM) requires a six-monthly update from all the pharmaceutical industry subcontractors, ranging from the synthesis of the active pharmaceutical ingredients (API) to the packaging process. Healthcare professionals are thus legitimately entitled to require such transparency from pharmaceutical companies.

Currently, only the European Medical Agency (EMA) has access to this information when it receives an application for a marketing authorization. This does not apply to antibiotics marketed before 2006. The figure below (Figure 2), which features in a report published by the NGO Changing Markets [28], illustrates the opacity of the antibiotic supply chain, highlighting the complexity of the production line as drugs transit through pharmaceutical plants in several countries to complete the different stages from API synthesis to final packaging.

### 2.2. The Lack of Environmental Criteria Imposed by the Buyers

Following several scandals, namely incriminating Chinese manufacturing plants, the EMA required subcontractors to follow Good Manufacturing Practices, which guarantee the quality of the products exported to European and American markets. However, no environmental criterion regarding the risk of bacterial resistance is required from the industry. While standards are imposed for discharging certain metals or toxic products, no measurement of antibiotics in the effluents nor searches for bacterial resistance are required. Such a criterion, an essential one in our view, do not feature among the aims of the regulatory authorities who monitor the quality and efficacy of medicines. This constitutes a major legal and regulatory gap and explains why subcontractors do not implement techniques known to improve the environmental impact of production, such as the internal treatment of wastewater according to the zero liquid discharge (ZLD) [29] process, theoretically installed in production plants, but without verification nor sanction in India [30]. Illegal discharge of industrial waste is also common [31].

While most studies show (experimentally or theoretically) a causal relationship between discharge into effluents and the spread of antibacterial resistance [32], the European Medical Agency has not yet legislated for this. Indeed, the EMA did not include the risk of antimicrobial resistance when updating its recommendations regarding the Environmental Risk Assessment (ERA—Draft of 30 November 2018), a mandatory requirement when applying for marketing authorization. Yet, threshold values have been published and could be used as benchmarks in this domain, particularly the PNEC-MIC (Predict No Effect Concentration), i.e., the antibiotic concentration in relation to the MIC, which would have no effect on the emergence of resistance genes [33,34].

Central procurement bodies that purchase antibiotics for our hospitals choose their suppliers according to quality, indication, and price. To this day, other criteria, such as the requirement for ecological production, are not considered during the transaction.

Systematically including a fixed percentage for these criteria could influence markets. For community-based consumers, clear information with an ecological score of the eco-score type could tip the balance towards a particular generic.

### 2.3. French and European Sovereignty

The Covid syndemic made us aware of our country’s lack of autonomy to protect and care for the French population; shortages of protective equipment, difficulties in producing and organising reliable tests, and a limited industrial capacity for drug production, made it impossible to produce vaccines rapidly and constrained the health care system, which suffered a shortage of antibiotics due to the lockdown in China [35,36]. Our country no longer has the means to meet the challenge of maintaining national sovereignty in terms of public health. This situation is likely to be shared by many other Western countries, and should be a wake-up call for us to reconsider the absolute necessity to control the entire range of the care process, from drugs, medical devices, and diagnostic tools to digital data and biological and virological logistics.

Although it is now crucial to realize how dependent we have become and to reverse the situation, the process will take time. Meanwhile, our country, in cooperation with European and other international partners, must require antibiotic producers to bring about a major change in their method of operating, which should be subjected to external audits.

Today, many scientists, industrialists, and politicians recognize this pollution as an unnecessary and unacceptable risk. However, any attempt to assess and reduce this risk quickly comes up against a complex reality. This requires an understanding of the incentives and disincentives via a systemic approach, as Nijsingh and Larsson have recently shown [37] (Figure 3).

During the Davos World Economic Forum in 2017, many industrialists committed to developing a roadmap to combat antimicrobial resistance, including R&D, access to drugs, antimicrobial stewardship, and environmental criteria. While such an initiative may be commended (AMR Industry Alliance), the environmental intentions are hardly convincing today. In 2020, some left the group because of the cost involved. The first progress report states that 76% of AMR Alliance members’ production sites adhere to the common framework for the manufacturing of antibiotics; however, the emission thresholds for the emergence of antibiotic resistance are still being evaluated and thus cannot be considered to support any commitment.

National and international political pressure should thus be enhanced by bringing together scientists and informing consumers via the media, raising the overall awareness of the risks involved in relation to this resistance in the context of the diminishing scope for developing new antibiotics.

Several European propositions were put forward at the end of 2021, and it is to be hoped that beyond supporting the industry and the development of new antimicrobial agents, these will stress the principles of restraint in the use of antibiotics, promote narrow-spectrum compounds, and advocate for the implementation of strict environmental measures [38,39,40,41]. A very recent document published by the pharmaceutical industry appears to express more transparent and responsible environmental concerns [42].

In France, the introduction of environmental criteria regarding the choice of antibiotics within a roadmap to counter antimicrobial resistance is currently being discussed.

To sum up, we must all play our part:

Information is of paramount importance (transparency, cooperation, and control), even more so as new players are arriving on the scene of antibiotic production in South America and Africa, where the same problems arise and similarly threaten the local populations.

Through coordinated action at the European and international levels, we must drastically reduce antibiotic use in livestock farming and in human medicine and improve antibiotic production from subcontractors. High-income consumer countries, public institutions, and industry are in a key position to initiate effective change through regulatory, economic, and political incentives. The ethical dimension of this major issue should be discussed as a priority.

Authorities must now consider measures to reduce the environmental impact of Antibiotics. There are numerous proposals:Review, if necessary, the manufacturing and supply provisions in pharmaceutical legislation to improve transparency and strengthen the oversight of the antibiotic production chain;Clarify all responsibilities to ensure overall environmental sustainability, preserving drug quality, and ensuring our readiness for new technologies;Strengthen the environmental criteria in the marketing authorization process;Increase the share of the environmental assessment in the scoring grids for public procurement tenders;Develop an eco-toxicity classification for the various antibiotic compounds that can be easily understood by health professionals, e.g., the PBT index;Based on the above, promote the use of the least eco-toxic antibiotics;Define discharge limits in line with current guidelines to combat antimicrobial resistance, e.g., PNEC-MIC;Regarding pharmaceutical dispensing, promote single-dose dispensing by community pharmacies, with prior information for patients and pharmacists, along with packaging that is adapted to this type of dispensing in public contracts.

## Figures and Tables

**Figure 1 antibiotics-11-00980-f001:**
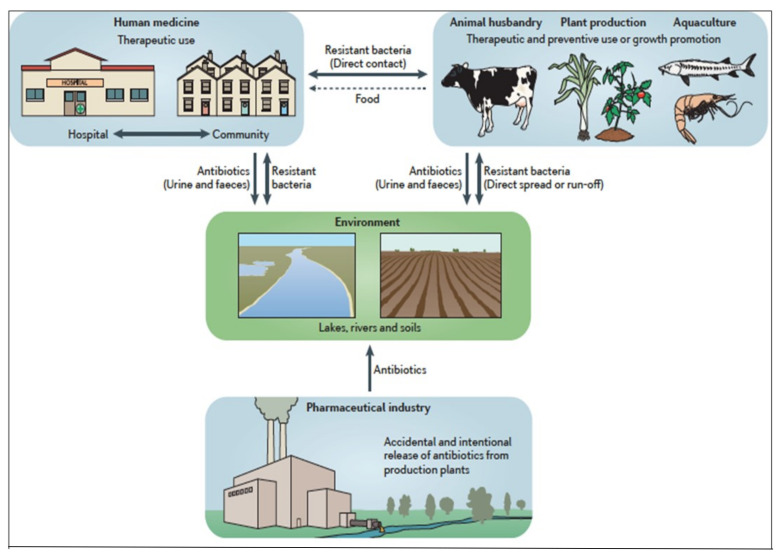
Ecology of antibiotic resistance. Reproduced with permission from Andersson et al., 2014 [1]. An overview of the ecological impact of antibiotics, sowing how these drugs are cycled between different environments, such as the medical environment, agricultural settings, the aquacultural environment, the pharmaceutical industry, and the wider environment. A large percentage of the antibiotics that are used globally (20–80% depending on the antibiotic class) are released in the environment in an active form, via the excretion of drugs in urine and faeces and the intentional or accidental release of drugs. Thus, antibiotics will exert selective pressure on bacteria in humans, animals and plants, owing to international use, and in the wider environment, owing to unintentional spill-over. This imposes a widespread selective pressure on bacteria, leading to the selection of resistant strains, which are also capable of transmitting between different environments, thereby creating the potential for the global movement of antibiotic-resistant genes and determinants [1].

**Figure 2 antibiotics-11-00980-f002:**
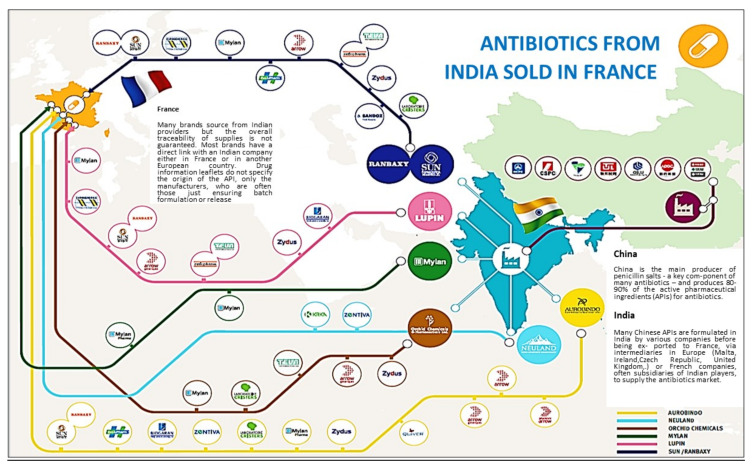
The complex circuits of antibiotic production. Reproduced and translated with permission from https://changingmarkets.org/portfolio/bad-medicine/ accessed on 16 May 2022 [28].

**Figure 3 antibiotics-11-00980-f003:**
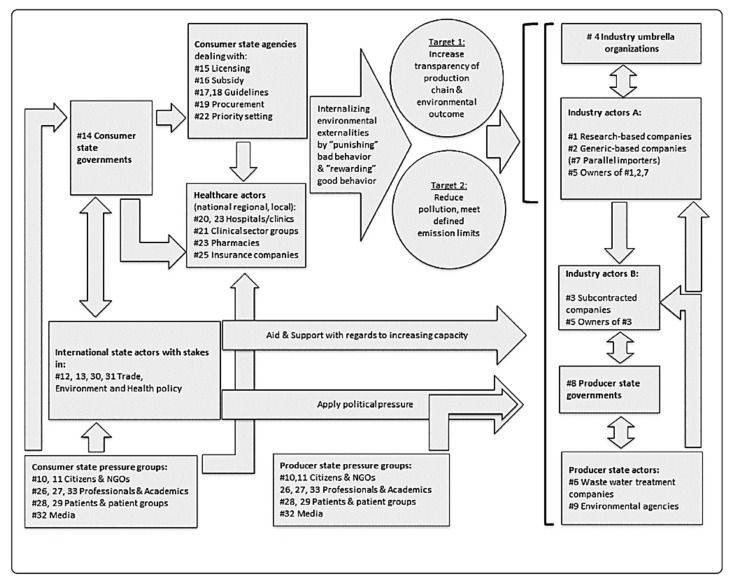
Examples of the chains of actions of different actor types to incentivize the action of other actors to improve the management of environmental–industrial antibiotic pollution. Reproduced with permission from Nijsingh, Niels et al. [37].

## Data Availability

Not applicable.

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
