# Peer review of "Advocacy for Responsible Antibiotic Production and Use"

_antibiotics, 2022, doi:10.3390/antibiotics11070980_

Round 1
Reviewer 1 Report
Dear authors
This is an interesting work that needs revision of the text.
Please consider some revisions in figures, for example figure 2- the explaining text of the figure should be in the same language as the body of the text of the paper. Even though this is an image reprinted with authorization, it should be adapted in order to have legends in English language, with adaptation under permission of the author.
Titles of figures and legends of figures, should be separated.
Figure 3, shows legend superposed to the image compromising the view.
Page 5 - What do you mean by “Covid syndemic”?
References need to be completed, reviewed and presented in an adequate system of bibliography.
Numbering of references referred in the text, should be uniform, some are in roman and other in decimal (for example in page 2, reference 10).
Page 7 - please consider changing Reference by References.
Author Response
Thank you for your review.
The following issues have been addressed as per your suggestions:
Titles of figures and legends of figures, should be separated. This has been corrected.
Figure 3, shows legend superposed to the image compromising the view. This has been corrected.
Page 5 - What do you mean by “Covid syndemic”? We have added two references on this subject (line 166).
References need to be completed, reviewed and presented in an adequate system of bibliography. Many references have been added, and are now presented in an adequate system of bibliography.
Numbering of references referred in the text, should be uniform, some are in roman and other in decimal (for example in page 2, reference 10). This has been corrected.
Page 7 - please consider changing Reference by References. This has been corrected.
Reviewer 2 Report
Brief Summary
Mondain et al. embark on a journey to describe the current challenges in antibiotic production and use. Specifically, they introduce the one health approach. Mondain et al. discuss how antibiotics use in animals and antibiotics manufacturing contribute to rising resistance to antibiotics locally and worldwide.
Significance
The discovery and use of antibiotics has improved both longevity and quality of life. The rise of antibiotics resistance threatens to erase the advancement of medicine and society made possible by antibiotics use. Understanding processes that contribute to rising resistance to antibiotics is an urgent need. Furthermore, this data is critical for the design of antibiotics manufacturing, use, and disposal in the future.
Recommendations:
I enthusiastically recommend accepting this paper with minor revisions for publication at the Antibiotics Journal. I am listing below minor suggestions for clarifying details described in this review. I am not recommending any additional experiments.
General:
- Please revise the references to journals guidelines. Including changing website links to PubMed to full authors list, journal name, and year of publication. Change reference system from roman letters to numbers.
“References: References must be numbered in order of appearance in the text (including table captions and figure legends) and listed individually at the end of the manuscript. We recommend preparing the references with a bibliography software package, such as EndNote, ReferenceManager or Zotero to avoid typing mistakes and duplicated references. We encourage citations to data, computer code and other citable research material. If available online, you may use reference style 9. below.”
https://www.mdpi.com/journal/antibiotics/instructions#preparation
- Please clearly number the subsections of the paper.
- Please add an English translation to figure 2, either within the figure, or in the legend.
- Please remove the reference text box from the figure 3 image, which is obstracting the lower left part of the figure, and place in legend.
- Many references are missing throughout the manuscript. Please revise and find reliable references in scientific literature or official reports by recognized institutions, to support the statements made throughout the paper.
In your abstract I suggest that you can use a citation more recent than the 2016 UN meeting. There are many statements and reports available by the UN and the WHO. For example, you could cite the “Call to Action on Antimicrobial Resistance (AMR) – 2021” by the UN in 2021 https://www.un.org/pga/75/wp-content/uploads/sites/100/2021/04/Call-to-Action-on-Antimicrobial-Resistance-AMR-2021.pdf
Another option is a recent WHO report, such as the “Global Antimicrobial Resistance and Use Surveillance System (GLASS) Report: 2021”
https://www.who.int/publications/i/item/9789240027336
In your first paragraph, could you please add a sentence to elaborate that antibiotic is added as a food supplement to animal food, on a daily basis. This would emphasize that antibiotic is not used as a medicine in case of bacterial infections in chickens and other animals. Readers might not be familiar with this practice of label and use of antibiotics as a food supplement, instead of as a medicine.
Can you add information about the regulations of antibiotic use in livestock in regions outside of Europe?
Comment at “with 90% of the population harboring such bacteria in their gut flora [xiii]”. Could you please elaborate and compare what is the abundance of harboring MDR bacterial diarrhea in the gut flora, in people in other regions?
Comment at “The Covid syndemic made us aware of our country’s lack of autonomy”. There is no reference to statements made about the COVID pandemic. Could you please add reference to this section?
Author Response
Thank you very much for your encourageing comments.Pleasefind hereunder our replies.
- Please revise the references to journals guidelines. Including changing website links to PubMed to full authors list, journal name, and year of publication. Change reference system from roman letters to numbers.
“References: References must be numbered in order of appearance in the text (including table captions and figure legends) and listed individually at the end of the manuscript. We recommend preparing the references with a bibliography software package, such as EndNote, ReferenceManager or Zotero to avoid typing mistakes and duplicated references. We encourage citations to data, computer code and other citable research material. If available online, you may use reference style 9. below.”
https://www.mdpi.com/journal/antibiotics/instructions#preparation
The format of the references has been corrected.
- Please clearly number the subsections of the paper. This has been done.
- Please add an English translation to figure 2, either within the figure, or in the legend. This has been done
- Please remove the reference text box from the figure 3 image, which is obstracting the lower left part of the figure, and place in legend. This has been removedand placed in the legend.
- Many references are missing throughout the manuscript. Please revise and find reliable references in scientific literature or official reports by recognized institutions, to support the statements made throughout the paper.
Several references have been added.
In your abstract I suggest that you can use a citation more recent than the 2016 UN meeting. There are many statements and reports available by the UN and the WHO. For example, you could cite the “Call to Action on Antimicrobial Resistance (AMR) – 2021” by the UN in 2021 https://www.un.org/pga/75/wp-content/uploads/sites/100/2021/04/Call-to-Action-on-Antimicrobial-Resistance-AMR-2021.pdf
Another option is a recent WHO report, such as the “Global Antimicrobial Resistance and Use Surveillance System (GLASS) Report: 2021”
https://www.who.int/publications/i/item/9789240027336
We have changed this as suggested.
In your first paragraph, could you please add a sentence to elaborate that antibiotic is added as a food supplement to animal food, on a daily basis. This would emphasize that antibiotic is not used as a medicine in case of bacterial infections in chickens and other animals. Readers might not be familiar with this practice of label and use of antibiotics as a food supplement, instead of as a medicine.
We have specified this in the relevant paragraph: lines 23-24. We have also added a sentence at the end of this paragraph with two references: lines 43-45, ref #12 and 13.
Can you add information about the regulations of antibiotic use in livestock in regions outside of Europe?
We have added a reference to a very comprehensive paper on the subject by Kirchhelle: line 23 (ref #3)
Comment at “with 90% of the population harboring such bacteria in their gut flora [xiii]”. Could you please elaborate and compare what is the abundance of harboring MDR bacterial diarrhea in the gut flora, in people in other regions?
The reference was not reliable enough, so this part of the sentence was removed. We have added other references (lines 66 to 71), one of which describes trends in carriage rates in different parts of the world (ref # 19, 20 and 21).
Comment at “The Covid syndemic made us aware of our country’s lack of autonomy”. There is no reference to statements made about the COVID pandemic. Could you please add reference to this section?
We could not find a scientific report on the subject but suggest the following article published on a national news programme:
https://www.france24.com/en/20200508-pandemic-disarmament-why-france-was-ready-for-covid-19-a-decade-too-soon